# A synaptic F-actin network controls otoferlin-dependent exocytosis in auditory inner hair cells

**Philippe FY Vincent[1]\*, Yohan Bouleau[1], Christine Petit[2,3,4,5,6], Didier Dulon[1,3]\***

[1]Bordeaux Neurocampus, Equipe Neurophysiologie de la Synapse Auditive, Université de Bordeaux, Bordeaux, France; [2]Unité de Génétique et Physiologie de l'Audition, Institut Pasteur, Paris, France; [3]UMRS 1120, Institut National de la Santé et de la Recherche Médicale (INSERM), Paris, France; [4]Sorbonne Universités, UPMC Université Paris, Paris, France; [5]Syndrome de Usher et Autres Atteintes Rétino-Cochléaires, Institut de la Vision, Paris, France; [6]Collège de France, Paris, France

**Abstract** We show that a cage-shaped F-actin network is essential for maintaining a tight spatial organization of Cav1.3 $Ca^{2+}$ channels at the synaptic ribbons of auditory inner hair cells. This F-actin network is also found to provide mechanosensitivity to the Cav1.3 channels when varying intracellular hydrostatic pressure. Furthermore, this F-actin mesh network attached to the synaptic ribbons directly influences the efficiency of otoferlin-dependent exocytosis and its sensitivity to intracellular hydrostatic pressure, independently of its action on the Cav1.3 channels. We propose a new mechanistic model for vesicle exocytosis in auditory hair cells where the rate of vesicle recruitment to the ribbons is directly controlled by a synaptic F-actin network and changes in intracellular hydrostatic pressure.

**\*For correspondence:** philippe.vincent@inserm.fr (PFV); didier.dulon@inserm.fr (DD)

**Competing interests:** The authors declare that no competing interests exist.

## Introduction

Auditory hair cells convert tiny variations of sound pressure through the displacement of their apical hair bundles into analogous voltage waveforms. Neural encoding of these microphonic potentials occurs at the ribbon synapses of inner hair cells (IHCs) by mechanisms involving Cav1.3 channels (*Platzer et al., 2000*; *Brandt et al., 2003*; *Brandt et al., 2005*) and otoferlin-dependent exocytosis of synaptic vesicles (*Roux et al., 2006*; *Beurg et al., 2010*, *Vincent et al., 2014*). Unlike most neuronal central synapses, IHCs have the extraordinary property to sustain indefatigably high rates of exocytosis during continuous sound stimulation (*Safieddine et al., 2012*). The precise molecular mechanisms underlying this fast and massive recruitment of synaptic vesicles to the IHC ribbon active zones still remain elusive. The implication of an unconventional molecular regulation of synaptic vesicle fusion and replenishment of the releasable pool of vesicles has been proposed (*Nouvian et al., 2011*; *Vogl et al., 2015*). In other secretory cells such as neuroendocrine cells, a network of sub-membranous cortical F-actin is known to influence exocytosis greatly by tightly regulating plasma membrane tension and the access of the granules to the secretory sites (*Apodaca, 2002*; *Torregrosa-Hetland et al., 2011*; *Gutierrez and Gil, 2011*). In central neuronal synapses, F-actin is also involved in maintaining vesicle pools and regulating vesicle mobility (*Cingolani and Goda, 2008*). Electron-tomography evidence for F-actin and microtubules near the synaptic ribbons has been observed in bullfrog hair cells (*Graydon et al 2011*). Whether IHC synaptic exocytosis is modulated by F-actin remains unknown. Other factors affecting membrane tension and exocytosis in many cell types, such as mast cells, include hydrostatic pressure across the membrane (*Solsona et al., 1998*). This latter factor has also been shown to influence the spontaneous

**eLife digest** To hear a sound, the pressure produced by sound waves must be converted into an electrical nerve signal. The cells inside the ear that perform this transformation are called hair cells, which are so named because they have hundreds of hair-like structures on their upper surface. Pressure from sound waves causes movements in the inner ear that bend these 'hairs'. This causes the hair cells to release chemical signals to neighboring nerve cell terminals that ultimately transmit information about the sound to the brain.

The chemical signals are stored inside the hair cells in bubble-like compartments called vesicles. To release the chemicals from the cell, the vesicles merge with the membrane that surrounds the hair cell. Most cells that communicate in this way are limited in how long they can transmit such messages. However, hair cells can continuously fuse vesicles to the membrane even when a sound lasts for a long time. This suggests that the hair cells have a different way of producing vesicles and getting them to the membrane than other cell types.

Inside the hair cells, vesicles are stored in regions called active zones. Each active zone contains a "ribbon" (attached to which are hundreds of vesicles) and also ion channels that allow calcium ions to flow into the cell. (An increase in calcium ion concentration inside the cell is necessary for the vesicle to fuse with the cell membrane and so release its chemical content). Now, Vincent et al. show that in hair cells, a cage-like network made from a protein called actin surrounds each active zone. This network helps to position the calcium ion channels. Treating the hair cells with a compound that disorganized the actin networks speed up the process of vesicle movement, which suggests that the actin network also controls the rate at which vesicles reach the membrane.

Next, it will be important to identify how the actin network interacts with other molecules that help vesicles to release their contents; in particular a protein called otoferlin, which is thought to act as a calcium ion sensor.

and evoked activity of vestibular hair cells of the dogfish by mechanisms that remain unknown (*Fraser and Shelmerdine, 2002*). In the present study, we investigated whether F-actin and intracellular hydrostatic pressure regulate synaptic exocytosis in mouse IHCs.

## Results and discussion

Direct labeling of F-actin with fluorescent phalloidin revealed the presence of a dense F-actin network that surrounded the IHC ribbon synaptic zones (*Figure 1A*). This network extended underneath the plasma membrane and formed intracellular dense cages beneath the synaptic ribbons. These cages displayed a mean size diameter of $0.8 \pm 0.1$ µm (n = 186 actives zones; *Figure 1B*). Overlapping with otoferlin, each F-actin cage was generally associated with one ribbon and one Cav1.3 co-immunoreactive patch. Similarly, $Ca^{2+}$ channels and the secretory machinery have been shown to be associated with the borders of F-actin cytoskeletal cages in chromaffin cells (*Torregrosa-Hetland et al., 2011*).

A 45-min treatment of the organ of Corti in vitro with 1 µM extracellular latrunculin-A completely disorganized the synaptic F-actin cages and the Cav1.3 immunoreactive-patches at the IHC ribbons (Figure 1C, D). The mean distance between the Cav1.3 immunoreactive patches and the ribbons increased from $218 \pm 11$ nm (n = 71) in controls to $260 \pm 12$ nm with latrunculin-A treatment (n = 102; p<0.05). In the latter conditions, surprisingly, whole-cell patch clamp recordings revealed a largely facilitated exocytosis as compared to controls, while voltage-gated $Ca^{2+}$ currents were unchanged (Figure 2A,B). After 100 ms depolarization, from -80 to -10 mV, the exocytotic response in control IHCs reached a maximum amplitude of $22.0 \pm 2.9$ fF (n = 11). Considering that a 40 nm diameter synaptic vesicle corresponds to 37 aF (*Lenzi et al., 1999*), we estimated a RRP size of about 590 vesicles, i.e 33 vesicles per ribbons if we assume a number of 18 ribbons per IHCs. This number of RRP vesicles fit well with previous findings (*Johnson et al., 2008*; *Vincent et al., 2014*). In latrunculin-treated IHCs, the exocytotic responses reached, at 100 ms, $32.4 \pm 3.3$ fF (n = 18), a value significantly larger as compared to control IHCs (p<0.05; *Figure 2B*). Remarkably, while the exocytotic response saturated at 90–100 ms in controls, the response did not show saturation in

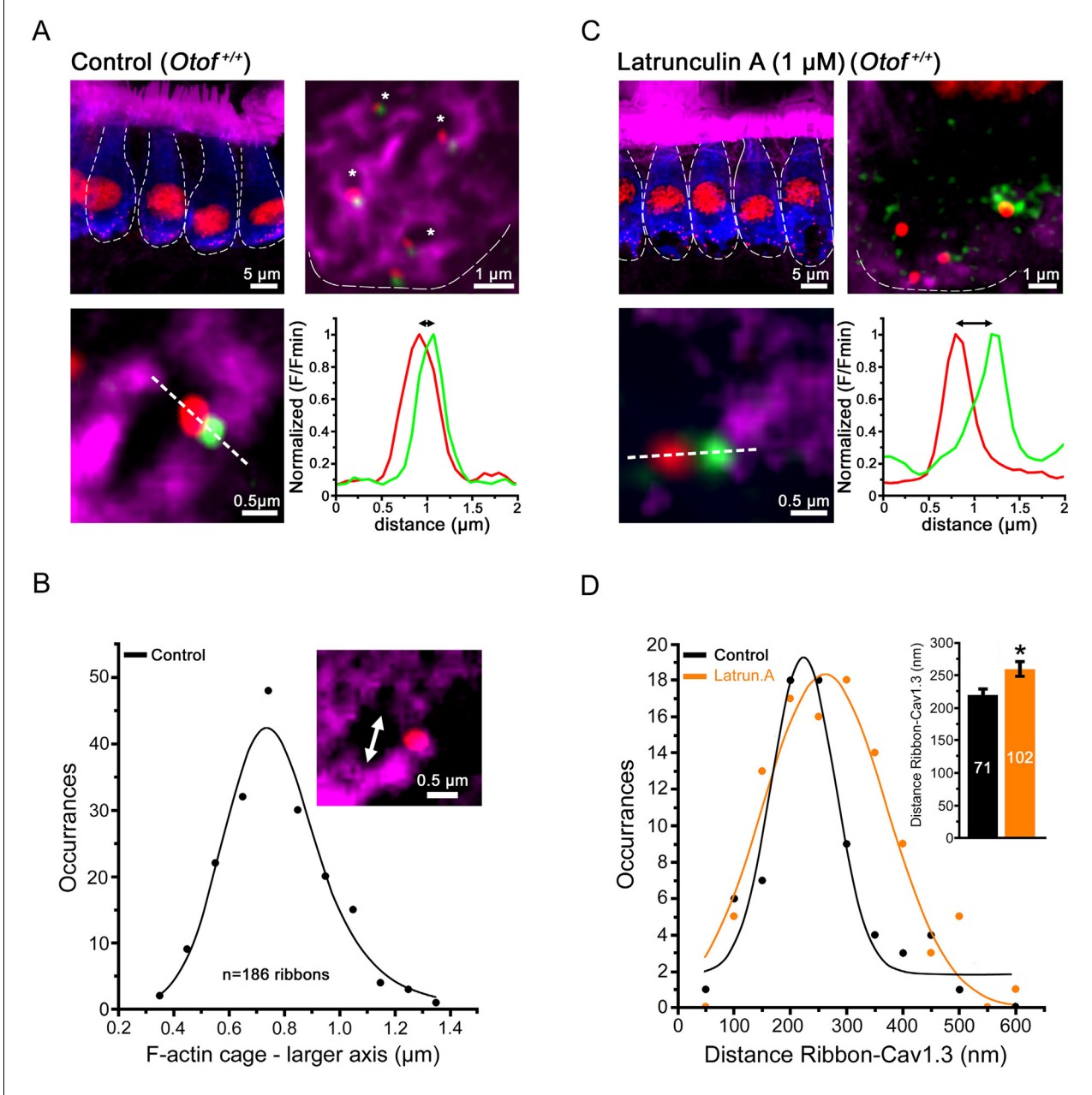

**Figure 1.** Confocal imaging of the synaptic F-actin cages in IHCs. (A) Confocal images from averaged Z-stack projection (20 slices of 0.25 μm) of P13-IHCs labeled in blue with otoferlin-immuno-reactivity. Directly visualized with fluorescent-phalloidin (purple), F-actin intensively labelled the cuticular plate and the stereocilia but also in a punctated manner the synaptic basal pole of the IHCs. In this latter area, at higher magnification (averaged Z-stack projection of 8 slices of 0.25 μm), the synaptic F-actin forms a mesh of cages (see right panel where the blue channel of otoferlin is omitted; the cages are indicated by the white asterisks). At each border of the synaptic F-actin cages was generally attached one synaptic ribbon (red) and one associated Cav1.3 patch (green) as indicated in the lower left panel. The graph represents an example of fluorescent intensity profile through the white dashed line crossing the ribbon and the associated Cav1.3. (B) The graph indicates the Gaussian distribution of the larger axis (double white arrow head) of each F-actin cage. (C) A 45 min treatment with extracellular latrunculin-A disrupted the synaptic F-actin cages. The black holes at the base of the IHCs likely indicated swollen IHC active zones produced by the synaptic F-actin disorganization. At higher magnification (right panel), note also the disorganization of the Cav1.3 clusters (green) at the ribbons, as indicated by a larger distance in their respective fluorescent intensity profile distribution (bottom graph). (D) Comparative Gaussian distribution of the center mass distance between Cav1.3 and ribbon in controls (black, n = 71 active zones) and latrunculin-treated (orange, n = 102 active zones) IHCs. The inset histogram indicates the mean ± SEM distance in both conditions. *p<0.05.

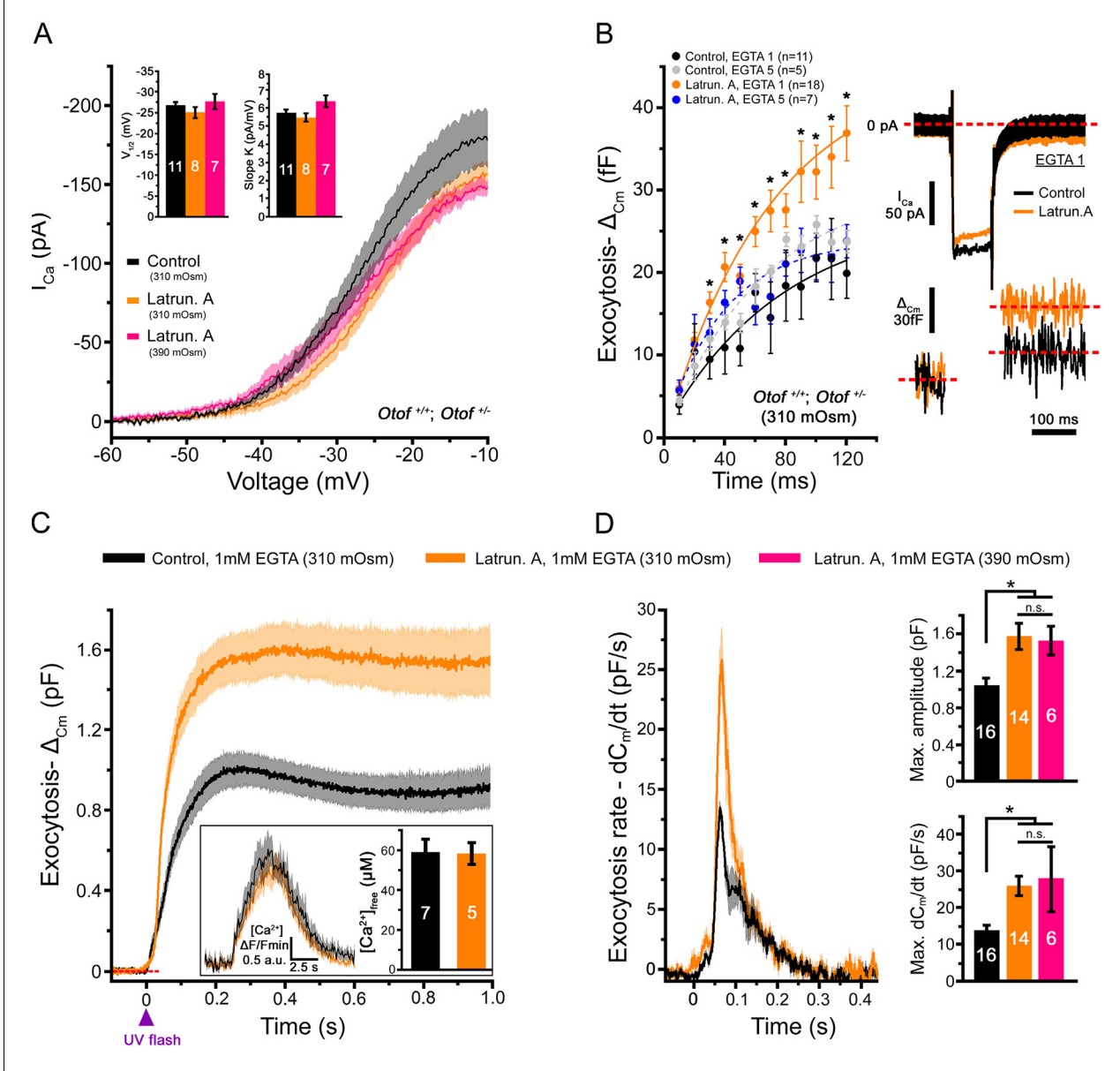

**Figure 2.** Latrunculin-A treatment facilitated exocytosis in IHCs. (A) $Ca^{2+}$ currents, evoked by a voltage-ramp protocol, were not significantly affected by latrunculin-A (orange). The parameters of the Boltzman fit are indicated by the histograms (mean ± SEM). Rising intracellular osmotic pressure from 310 mOsm to 390 mOsm did not affect the $Ca^{2+}$ currents in latrunculin-treated IHCs (pink). For each curves, the darker line indicates the mean responses and the light filled area the standard error. (B) Exocytosis evoked by voltage-steps from -80 mV to -10 mV was largely facilitated after latrunculin-A treatment. At right, examples of $Ca^{2+}$ currents and capacitance jumps ($\triangle_{Cm}$) are shown for a control (black traces) and latruncutin-treated IHC (orange traces). In 5 mM intracellular EGTA condition, the facilitation was greatly reduced (blue points). Data are represented as mean ± SEM. *p<0.05. (C) Exocytosis under $Ca^{2+}$ uncaging was also facilitated after latrunculin-A treatment. The jump in the concentration of intracellular free $Ca^{2+}$ was similar in control and latrunculin-treated IHCs (inset; p = 0.9). (D) Left, comparative exocytotic rates of control and latrunculin-treated IHCs obtained from the first derivative ($dC_m/dt$) of the curves in C. Right, comparative maximum exocytotic amplitude and peak rate histograms (mean ± SEM). Increasing intracellular hydrostatic pressure from 310 to 390 mOsm (pink bars) did not affect exocytosis in latrunculin-treated IHCs. Numbers of cells are indicated in the histogram. *p<0.05 and n.s. as non significative.

latrunculin-treated IHCs. Since the exocytotic response was unchanged for short impulses below 30 ms (*Figure 2B*), these results suggested that the disruption of F-actin did not affect the steps of vesicle docking and priming but facilitated the replenishment of the RRP. In bassoon mutants with

abnormal number of anchored ribbons and reduced $Ca^{2+}$ currents both short (20 ms) and long (100 ms) impulse activated-exocytosis were affected (*Jing et al., 2013*).

The exocytotic facilitation in latrunculin-treated IHCs was greatly reduced to $21.8 \pm 2.5$ fF at 100 ms (n = 7) and comparable to controls (p=0.9) when intracellular $Ca^{2+}$ buffering was increased with 5 mM EGTA (*Figure 2B*). This sensitivity to EGTA suggested a spatial disorganization of the $Ca^{2+}$ channel clusters in regards to the release sites in latrunculin-treated IHCs, in good agreement with confocal imaging (Figure 1C,D). Rising the EGTA concentration could also affect the extent to which neighboring calcium sources interact and summate to produce a global effect on free calcium.

The intriguing question now is: why is the disruption of the synaptic F-actin with latrunculin-A facilitating exocytosis in IHCs? One explanation is that the synaptic F-actin network, in addition to organizing $Ca^{2+}$ microdomains, also acts as a diffusion barrier for synaptic vesicles limiting the access to the site of release, as suggested in some central synapses (*Cingolani and Goda, 2008*). Its disruption with depolymerising agents would therefore facilitate vesicle replenishment of the release sites, as shown in a large variety of secretory cells (*Malacombe et al., 2006*), by increasing the number of available vesicles for docking and priming. Alternatively, a disrupted F-actin could facilitate the diffusion of $Ca^{2+}$ from its sites of entry and stimulate replenishment.

To directly test these hypothesis, we studied the effect of F-actin depolimerization on exocytosis triggered by intracellular $Ca^{2+}$ uncaging, i.e. independently of the activation and the organization of the $Ca^{2+}$ channels. In these experiments, a large exocytotic facilitation was also observed in latrunculin-treated IHCs (Figure 2C,D). The peak exocytotic rate, obtained by measuring the first derivative function of the curves in *Figure 2C*, was nearly two fold larger in latrunculin-treated IHCs as compared to controls (*Figure 2D*; $25.9 \pm 2.7$ pF/s (n = 14) and $13.7 \pm 1.7$ pF/s (n = 16), respectively; p<0.05). The levels of the peak intracellular $Ca^{2+}$ concentration ($[Ca^{2+}]_{free}$) reached upon UV-flash $Ca^{2+}$ uncaging were verified to be similar in control (n = 7) and latrunculin-treated IHCs (n = 5), respectively $59 \pm 7$ μM and $57 \pm 5$ μM (p=0.9; Figure 2C-inset). These $Ca^{2+}$ uncaging experiments again suggested that a synaptic F-actin network controls the diffusion rate of the synaptic vesicles to the sites of release in IHCs.

Furthermore, since intracellular hydrostatic pressure has been suggested to influence membrane tension and exocytosis through the F-actin network in many cell types such as mast cells (*Solsona et al., 1998*), we probed $Ca^{2+}$-evoked exocytosis under various intracellular osmotic pressures in auditory IHCs. We first found that increasing intracellular osmotic pressure from 310 to 390 mOsm with sucrose produced a significant increase in the resting membrane capacitance of IHCs. The resting size of IHCs, voltage-clamped at -80 mV for a period of 2 min after break-in, was $9.93 \pm 0.27$ pF (n = 11, 310 mOsm) and $11.10 \pm 0.34$ pF (n = 10, 390 mOsm; p<0.05), respectively. This augmentation of the IHC resting membrane capacitance was about 50 times larger than the size of the RRP evoked by membrane depolarization (Figure 2B; RRP = 22 fF). Where does this large addition of membrane come from? One possible explanation was that high intracellular hydrostatic pressure triggers the fusion of a large amount of extrasynaptic vesicles to the plasma membrane, as previously suggested for $Ca^{2+}$ uncaging (*Vincent et al., 2014*).

Remarkably, in these latter conditions of intracellular hyperosmotic stress at 390 mOsm, voltage-dependent $Ca^{2+}$ currents displayed larger amplitude (Figure 3A,C) and accelerated activation kinetics as compared to control conditions at 310 mOsm (*Figure 3E*). A shift in their voltage-dependence toward more negative potentials was also observed (Figure 3A,D). The hydrostatic pressure effects on $Ca^{2+}$ currents were no longer visible when IHCs were pre-treated with latrunculin-A (*Figure 2A*, pink line) and were greatly reduced in IHCs lacking otoferlin ($Otof^{-/-}$; Figure 3B,C). This indicated that the Cav1.3 channels of IHCs are mechanosensitive, like the Cav1.2 channels in smooth muscle cells (*Lyford et al., 2002*). We cannot exclude the addition of $Ca^{2+}$ channels to the plasma membrane during the massive vesicular fusion caused by increased hydrostatic pressure. However, this mechanism appears unlikely in regards to results obtained in non-secretory cells such as smooth muscle and HEK cells where similar effect on $Ca^{2+}$ channels were obtained (*Lyford et al., 2002*). In our study, the sensitivity to membrane tension in IHCs required an intact synaptic F-actin network and otoferlin, a proposed synaptic $Ca^{2+}$ sensor thought to interact physically with the Cav1.3 II-III loop (*Ramakrishnan et al., 2009*). Notably, an increase in $Ca^{2+}$ current amplitude in low bath hydrostatic pressure (equivalent to increasing intracellular pressure) was also reported in dissociated guinea pig vestibular hair cells (*Duong Dinh et al., 2009*; *Haasler et al., 2009*). In these latter studies, the change in $Ca^{2+}$ currents could be interpreted as due to a pressure activation of $K^+$ currents

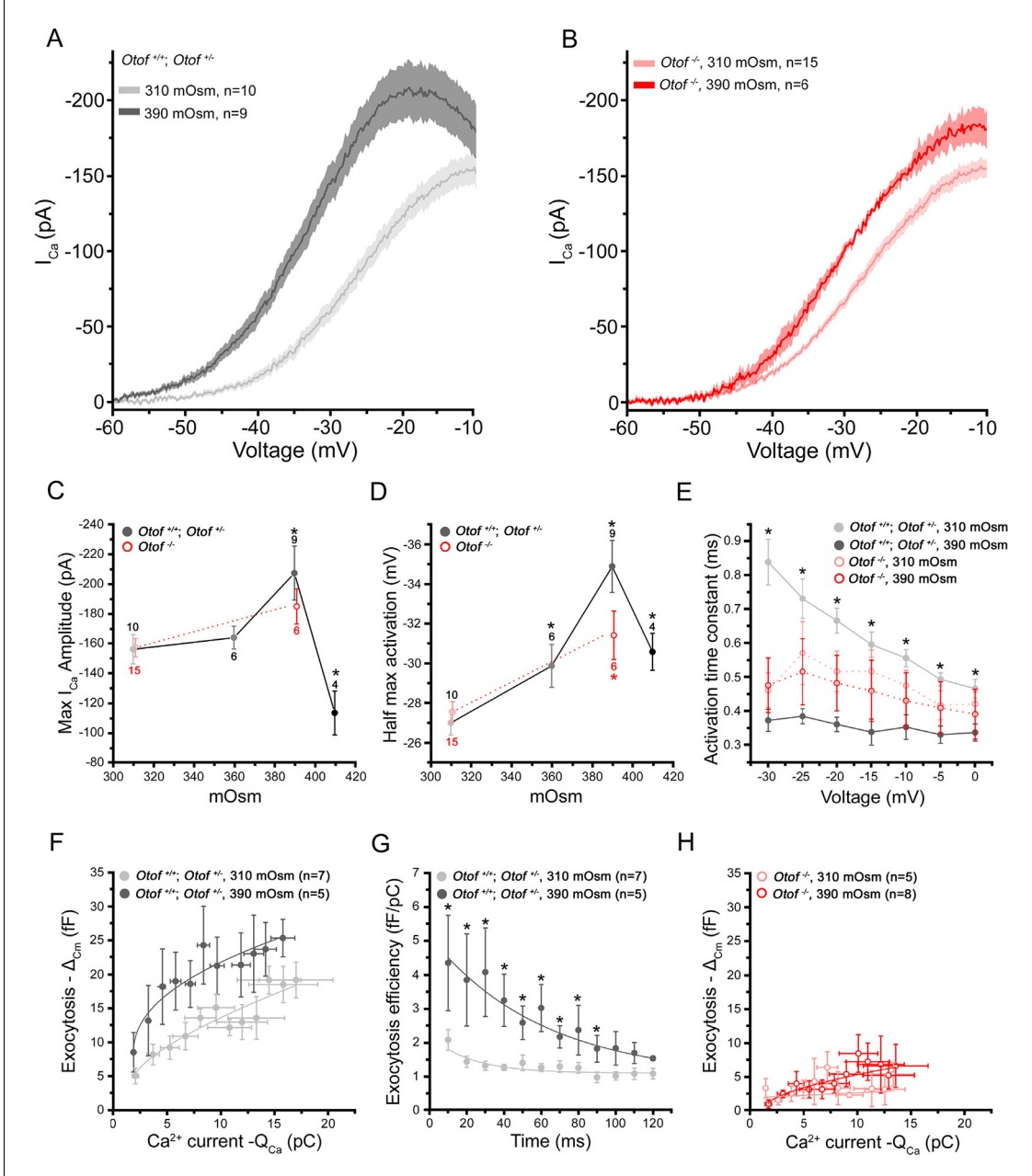

**Figure 3.** Intracellular hydrostatic pressure modulates $Ca^{2+}$ currents and exocytosis. (A) Comparative $Ca^{2+}$ currents evoked by a voltage-ramp protocol in $Otof^{+/-}$ and $Otof^{+/+}$ IHCs (expressing otoferlin). Cells were recorded with intracellular osmotic pressure at 310 mOsm (n = 10) or at 390 mOsm (n = 9). The darker line indicates the mean responses and the light filled area the standard error. (B) $Ca^{2+}$ currents in $Otof$-/- IHCs (lacking otoferlin) recorded in conditions similar to A. (C-D) Both maximum amplitude and half-maximum voltage-activation of $Ca^{2+}$ currents were maximally affected at 390 mOsm. Note that the slopes of the Boltzman fit of the activation curve in A and B were found slightly affected in IHCs expressing otoferlin (310 mOsm: K = 5.90 ± 0.14 pA/mV, n = 10 and 390 mOsm: K = 5.36 ± 0,09 pA/mV, n = 9; p < 0.05) but not in $Otof$-/-IHCs (310 mOsm: K= 6.18 ± 0.20 pA/mV, n=6 and 390 mOsm: K= 5.89 ± 0.28, n = 6; p = 0.71) (data not shown). Data are represented as mean ± SEM, with the number of cells indicated above each point and (*) indicating p < 0.05. (E) Activation kinetics of $Ca^{2+}$ currents, evoked by voltage-steps from -80 mV to different voltage levels, were significantly faster at 390 mOsm in hair cells expressing otoferlin but not in $Otof$-/- IHCs. For unknown reason the activation kinetics were faster in $Otof$-/-IHCs as compared to IHCs expressing otoferlin. Data are expressed as mean ± SEM. * p < 0.05. The number of cell is similar to D. (F) $Ca^{2+}$ efficiency of exocytosis was plotted as the change in membrane capacitance ($\Delta_{Cm}$) against the integral of the calcium current ($Q_{Ca}$) when depolarizing IHCs at a constant voltage-step from -80 to -10 mV with increasing duration from 10 to 120 ms. Data points at 310 mOsm and 390 mOsm were fitted with a power function with a $Ca^{2+}$ efficiency slope $A$ = 2.47 ± 0.45 fF/pC and 9.30 ± 2.5 fF/pC (p < 0.05) and a (power cooperative index) = 0.33 and 0.51 (p = 0.52), respectively. The supralinear power cooperative index of 0.3 and 0.51 found here was somewhat lower to previous values reported by **Cho et al. (2011)** and **Johnson et al (2005)**. (G) The exocytotic efficiency of each data point in F ($\Delta_{Cm}/Q_{Ca}$) was plotted for each depolarizing time at 310 mOsm

*Figure 3. continued on next page*

*Figure 3. Continued*

and 390 mOsm. Data were fitted with an exponential function with 22.2 ± 10.4 ms and 50.1 ± 11.9 ms, respectively (p < 0.05). **(H)** $Ca^{2+}$ efficiency of exocytosis (recorded in conditions similar to F) was unaffected when rising osmotic pressure from 310 to 390 mOsm in *Otof-/-*IHCs. Data were best fitted with a linear function with a similar slope of 0.12 ± 0.1 fF/pC and 0.48 ± 0.1 fF/pC (p = 0.2) at 310 and 390 mOsm, respectively. Data are represented as mean ± SEM. * p < 0.05.

leading to less pronounced depolarization. Although pressure effects on $K^+$ currents have also been shown in guinea-pig IHCs (*Kimitsuki, 2013*), we don't think that these currents would influence our $Ca^{2+}$ current measurements since we were working in conditions where most $K^+$ currents are blocked. Overall, our results suggested here that the mechanosensitivity of the Cav1.3 channels is mediated through an intact synaptic F-actin network.

Exocytosis triggered by voltage-activation of $Ca^{2+}$ channels from -80 to -10 mV (a voltage at which $Ca^{2+}$ current amplitude is maximum, *Figure 3A*) also showed maximum facilitation when intracellular pressure was increased from 310 to 390 mOsm (Figure 3F,G). Notably, on the contrary to what observed when disrupting the synaptic F-actin, exocytotis increased for short impulses (10–20 ms) at 390 mOsm (Figure 3F,G), suggesting that the last steps of vesicle exocytosis (docking, priming and/or fusion) were here accelerated. Exocytosis for longer steps up to 90 ms also increased, from 1.40 ± 0.23 fF/pC (n = 7, 310 mOsm) to 2.58 ± 0.49 fF/pC (n = 5, 390 mOsm; p<0.05) at 50 ms (*Figure 3G*). These latter results suggested that the replenishment rate of the RRP was also accelerated.

This facilitation of exocytosis, under intracellular hyperosmotic stress, appeared unrelated to the effects on $Ca^{2+}$ channels since it was also observed when directly uncaging intracellular $Ca^{2+}$ (*Figure 4*). The levels of the peak $[Ca^{2+}]_{free}$ reached upon UV-flash $Ca^{2+}$ uncaging were verified to be similar in 310 mOsm (n = 7) and 390 mOsm (n = 5) conditions, 59 ± 7 μM and 64 ± 6 μM (p=0.6; Figure 4A-inset). In these latter conditions of intracellular hyperosmotic stress, the strength of exocytotic facilitation was maximal at 390 mOsm and abruptly decreased at 410 mOsm, a high pressure at which the intrinsic organization of the IHC exocytotic machinery may have been damaged. At 390 mOsm, the maximum amplitude of the exocytotic response was 1.5 ± 0.15 pF (n = 9) as compared to 1.0 ± 0.1 pF (n = 16) in controls at 310 mOsm (p<0.05; Figure 4A,C). The peak exocytotic rate was largely increased at 390 mOsm as compared to 310 mOsm (25.0 ± 4.5 pF/s (n = 9) and 13.7 ± 1.7 pF/s (n = 16), respectively; p<0.05, *Figure 4D*). Interestingly, the facilitation of exocytosis by high intracellular hydrostatic pressure was not observed in IHCs treated with latrunculin-A (*Figure 2D*, pink bars) and in *Otof^-/-* IHCs (*Figure 3H* and Figure 4B–D). The peak $[Ca^{2+}]_{free}$ reached upon UV-flash $Ca^{2+}$ uncaging was unchanged in *Otof^-/-* IHCs as compared to control IHCs (*Vincent et al., 2014*).

Overall, the facilitation of exocytosis by high intracellular hydrostatic pressure could be explained by an increased membrane tension that impacts on membrane fusion (*Kozlov and Chernomordik, 2015*) but also by an increased vesicular mobility, possibly as the result of reduced molecular crowding and loosened vesicle interactions, accelerating the replenishment rate. In the same way, the mobility of vesicles in pancreatic cells and primary hepatocytes was shown to be affected by hydrostatic pressure, a process related to molecular crowding and microfilaments polymerization (*Nunes et al., 2015*).

Do large changes in hydrostatic pressure occur in the cochlea during physiological or pathological conditions? In the cochlea, large intercompartmental osmotic gradients from 289 to 322 mOsmol/kg $H_2O$ are present between the perilymphatic and endolymphatic compartments, respectively (*Sterkers et al., 1984*). These osmotic gradients are likely regulated by aquaporins present in outer sulcus cells (*Eckhard et al., 2015*) and genetic deletion of aquaporin-4 in mice leads to impaired hearing (*Li and Verkman, 2001*). Osmolarity changes in these inner fluid compartments have long been suspected to be contributing factors to inner ear disorders such as tinnitus and fluctuating hearing loss, inclusive of Menière's disease (*Angelborg et al., 1982*). The outer hair cells, producing the electro-mechanical amplification of sound in the cochlea, have been shown to be mechanically sensitive to extracellular osmotic variation (*Dulon, et al., 1987*), a factor that greatly influence hearing (*Choi and Oghalai, 2008*). Our study, showing that IHC exocytosis is sensitive to osmotic forces,

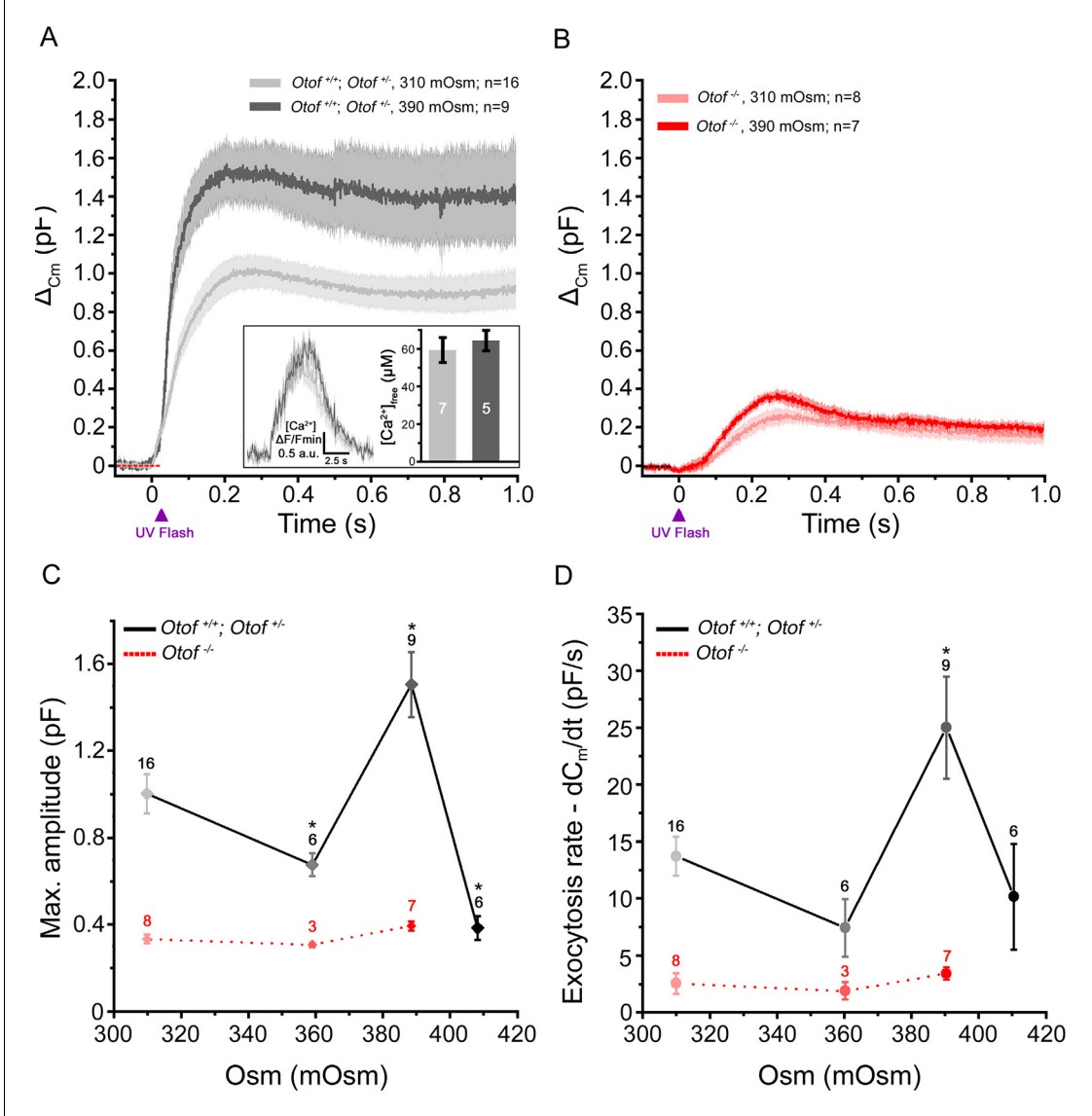

**Figure 4.** Exocytosis triggered by $Ca^{2+}$ uncaging is sensitive to intracellular hydrostatic pressure. **(A)** Exocytosis in IHCs expressing otoferlin was largely potentiated when rising osmotic pressure from 310 mOsm (light grey) to 390 mOsm (dark grey). The darker line in each condition indicates the mean responses and light filled area the standard error. The jump in the concentration of intracellular free $Ca^{2+}$ was similar in 310 and 390 mOsm conditions (inset; p = 0.6). **(B)** Exocytosis evoked in *Otof-/-* IHCs in conditions similar to A. **(C-D)** Comparative maximum amplitude and peak exocytotic rate at various intracellular osmotic pressure. The results showed a maximum facilitation at 390 mOsm. These pressure effects were not seen on the residual slow exocytosis of *Otof -/-* IHCs. Data points are means ± SEM. The number of IHCs is indicated above each point. * p < 0.05.

suggests that a defect in synaptic transmission at the auditory ribbon synapses may also underlie some hearing disorders associated with Menière's disease.

In conclusion, we propose that a synaptic F-actin network tightly controls the flow of synaptic vesicles during exocytosis at the IHC ribbons. This synaptic F-actin network also influences the sensitivity of exocytosis to hydrostatic pressure changes, presumably through a regulation of membrane tension at the active zones. Such actin-based regulation of intracellular hydrostatic pressure and membrane tension, recently shown in a variety of physiological processes such as in the secretory properties of neuroendocrine cells (*de Wit, 2010*; *Gutiérrez and Gil, 2011*), the mitotic cell rounding during cell division (*Stewart et al., 2011*), the photomechanical responses of photoreceptors (*Hardie and Franze, 2012*) to name a few, is here described for the first time in auditory hair cells.

## Materials and methods

### Tissue preparation

Experiments were performed in accordance with the guidelines of the Animal Care Committee of the European Communities Council Directive (86/609/EEC) and were approved by the ethics committee of the University of Bordeaux (animal facility agreement number C33-063-075). All mice (C57BL6 of either sex) were anesthetized by intraperitoneal injection of xylazin (6 mg/ml) and ketamine (80 mg/ml) mixture (Sigma Aldrich, St Louis, USA) diluted in physiological saline. The organs of Corti were dissected as previously described (*Vincent et al., 2014*).

Electrophysiological recordings were obtained from littermate *Otof*$^{+/+}$, *Otof*$^{+/-}$ or from knock-out (KO) otoferlin (*Otof*$^{-/-}$) C57BL6 mice at postnatal day 12–16 (P12-P16) inner hair cells (IHCs) in whole-mount organs of Corti in the apical cochlear area coding for frequencies ranging from 8 to 16 kHz.

The organ of Corti (OC) was freshly dissected under binocular microscopy in an extracellular solution maintained at 4°C containing (in mM): NaCl 135; KCl 5.8; CaCl$_2$ 1.3; MgCl$_2$ 0.9; NaH$_2$PO$_4$ 0.7; Glucose 5.6; Na pyruvate 2; HEPES 10, pH 7.4, 305 mOsm. The tectorial membrane was carefully removed and the OC was placed in a recording chamber and visualized under a 60x water immersion objective (CFI Fluor 60X W NIR, WD = 2.0 mm, NA = 1) attached to an upright Nikon FN1 microscope. The extracellular solution was complemented with 0.5 µM of apamin (Latoxan, Valence, France) and 0.2 µM of XE-991 (Tocris Bioscience, Lille, France) to block SK channels and KCNQ4 channels, respectively. All Ca$^{2+}$ current and capacitance recordings were performed in the presence of 5 mM extracellular Ca$^{2+}$ and carried out at room temperature (20–24°C).

### Whole cell recording and capacitance measurement

All patch clamp experiments were performed with an EPC10 amplifier controlled by pulse software Patchmaster (HEKA Elektronik, Germany). Patch pipettes were pulled with a micropipette Puller P-97 Flaming/Brown (Sutter Instrument, Novato, CA, USA) and fire-polished with a Micro forge MF-830, (Narishige, Japan) to obtain a resistance range from 3 to 5 MΩ. Patch pipettes were filled with an intracellular cesium-based solution containing (in mM): CsCl 145; MgCl$_2$ 1; HEPES 5; EGTA 1; TEA 20; ATP 2 and GTP 0.3; pH 7.2, 310 mOsm. To increase intracellular osmotic pressure, the cesium-based solution was complemented with different concentrations of sucrose (17g/L for 360 mOsm; 27g/L for 390 mOsm and 34 g/L for 410 mOsm).

Current-voltage (I-V) curves were recorded using two different protocols. First, cells were maintained at -80 mV and depolarized with a ramp protocol (1 mV/ms) from -90 mV to +30 mV in 120 ms. The voltage parameters (half max activation potential and the slope) of the IV curves were given by fitting the IV curves from -70 mV to -10 mV with a Boltzmann sigmoidal function:

$$y = \frac{A1 - A2}{1 + \exp^{(V - V_{1/2})/K}} \tag{1}$$

Where *A1* and *A2* are the minimum and the maximum y value. *V* is the voltage value and *V$_{1/2}$* is the half max voltage activation, *K* is the slope of the IV curve. Second, cells were step-depolarized from -80 mV to -5mV in 5mV increments for a constant time duration of 100 ms. Activation kinetics of the Ca$^{2+}$ currents were determined at different potentials with the latter protocol by using a single exponential fit:

$$y = y_0 + A1\exp^{(-x/t)} \tag{2}$$

Where *y$_o$* is the offset of activation, A1 the amplitude of the Ca$^{2+}$ current and *t* the time constant.

Real-time capacitance measurements (C$_m$) were performed using the Lock-in amplifier Patchmaster software (HEKA) by applying a 1 kHz command sine wave (amplitude 20 mV) at holding potential (-80 mV) before and after the pulse experiment, as previously described (*Vincent et al., 2014*). The time interval between each depolarization was set at 10 seconds to allow full replenishment of the RRP. The exocytosis efficiency was determined by fitting the data points with a power function:

$$\Delta_{Cm} = y_0 + A[x - x_c]^a \tag{3}$$

Where *y$_o$* is the initial value (fF), *A* the Ca$^{2+}$ efficiency slope (fF/pC), *a* the power cooperative index, *x$_c$* the Ca$^{2+}$ charge (Q$_{Ca}$ in pC) threshold of the response and *x* (pC) the Q$_{Ca}$ value for each

stimulation. The relationship between the $Ca^{2+}$ efficiency and the time was fitted with a exponential function:

$$y = y_0 + A1e^{-(x-x_0)/t} \tag{4}$$

Where $y_o$ et $x_0$ are the y and x offset, $A1$ the amplitude and $t$ the time constant.

Only cells with stable membrane resistance ($R_m$), leak current below 50 pA at holding potential (-80 mV) and stable series resistance below 15 MΩ were considered in the study. All $Ca^{2+}$ currents were leak-subtracted.

## Caged $Ca^{2+}$ photolysis

To trigger a fast increase in intracellular $Ca^{2+}$ concentration from the caged $Ca^{2+}$ chelator DM-nitrophen (Interchim, France), we used 200 ms brief flashes from a UV LED light source (Mic-LED 365, 128mW, Prizmatix, Givat Shmuel, Israel). The UV LED was directly connected to the epi-illumination port at the rear of our upright Nikon FN1 microscope and illumination was focalized through the 60X objective (CFI Fluor 60X W NIR, WD = 2.0 mm, NA=1). Hair cells were loaded with in mM, CsCl 145; HEPES 5; TEA 20; DM-nitrophen 10; $CaCl_2$ 10. In some experiments, in order to check that similar $Ca^{2+}$ release occurs in all our conditions, we added 50 µM OGB-5N in the intracellular solution. To determine the intracellular $Ca^{2+}$ concentration ($[Ca^{2+}]_{free}$), we used the following formula:

$$[Ca^{2+}]_{free} = K_D \times \frac{(F - F\mathrm{min})}{(F\mathrm{max} - F)} \tag{5}$$

Where $[Ca^{2+}]$ free is the caged $Ca^{2+}$ released by the UV flash, $K_D$ the constant dissociation of OGB-5N. We used the value of 23.3 µM determined by our previous *in vivo* calibration (*Vincent et al., 2014*). F the over time fluorescence, $F_{min}$ the minimum fluorescence and $F_{max}$ the maximal fluorescence. Fluorescence signals were plotted and compared as $\Delta F/Fmin$ where $\Delta F$ is equal to: F-Fmin. To vary the intracellular osmotic pressure from 310 to 360, 390 or 410 mOsm, the DM-nitrophen solution was complemented with sucrose as described above.

## Immunohistofluorescence

P13 organs of Corti (OC) were rapidly fixed with 100% methanol at -20°C for 30 min and washed with cold phosphate buffered saline (PBS). The tectorial membrane was carefully removed. Tissues were first incubated with PBS containing 30% normal horse serum for 1 hr at room temperature (RT). Then they were incubated with primary antibodies diluted with PBS (1:200) containing 5% horse serum and 0.1% triton X-100. The organization of the sub-membranous actin-F network was visualized using Phalloidin Fluoprobe 405 (1:100, Interchim, Montlucon, France; cat # FP-CA9870). Synaptic ribbons (CtBP2), Cav1.3 channels and otoferlin were simultaneously labeled with anti-CtBP2 (Goat polyclonal, Santa Cruz, USA; cat # SC-5966), anti-Cav1.3 (Rabbit polyclonal, Alomone labs, Jerusalem, Israel; cat # ACC-005) and anti-otoferlin (Mouse monoclonal, Abcam, Paris, France; cat # ab53233) antibodies, respectively. The organ of Corti was then washed with PBS and incubated in two steps with secondary antibodies at 1:500: first, Donkey anti-Goat Fluoprobe 547H (Interchim; cat # FP-SB2110) and Donkey anti-Mouse Fluoprobe 647 (Interchim; cat # FP-SC4110), second, after a PBS rinse with Goat anti-Rabbit Alexa Fluor 488 (Invitrogen; cat # A-11008). To disrupt F-actin, the organs of Corti were incubated prior to fixation with 1 µM latrunculin-A (Interchim, France; cat # FP-47143A ) for 45 min at RT.

Confocal imaging was performed with a Leica SP8 confocal laser-scanning microscope with a 63X oil immersion objective (NA = 1.4) and white light laser (470 to 670 nm) (Bordeaux Imaging Center). Phalloidin was imaged by using a diode laser at 405 nm also mounted on the microscope.

Stack images were acquired with the following parameters: laser power 40%, scan rate 700 Hz, scans averaged per XY section 4 times, step size 250 nm, pixel size 80 nm giving an X-Y image size of 41 x 41 µm (512 x 512 pixels). Images were then processed for three-dimensional (3D) blind deconvolution with AutoQuant X2 (MediaCybernetics). After deconvolution, images were processed with ImageJ software (W.S. Rasband, NIH, Bethesda, USA). Distance measurements between CtBP2 (ribbon) and the Cav1.3 channels were performed using the Image J software after 3D blind

deconvolution, as previously described (*Vincent et al., 2014*). The plot profile tool of Image J was used to determine the mean distance between the ribbon and the Cav.3 patch (center mass distance).

## Statistical analysis

Electrophysiological results were analyzed with Patchmaster (HEKA Elektronik, Germany), OriginPRO 9.1 (OriginLab, Northampton, USA) and IgorPro 6.3 (Wavemetrics, Oregon, USA). Results are expressed as mean ± SEM. Statistical analyses were performed by using the non-parametric Mann-Whitney U test with OriginPRO 9.1 software. The number of samples for each condition was indicated on graphs or in legends. The limit of significance was set at 0.05 ($p < 0.05$). When the statistical tests were found to be non-significant, the p value was given.

## Acknowledgments

We thank the Bordeaux Imaging Center for providing cutting-edge instruments for high resolution confocal microscopy and Professor Ray Cooke for copyediting the manuscript. This work was supported by the French state program ''Investissements d'Avenir'' (ANR-10-LABX-65; to C.P.).

## Additional information

### Funding

| Funder | Grant reference number | Author |
| --- | --- | --- |
| Agence Nationale de la Recherche | ANR-10-LABX-65 | Christine Petit |

The funders had no role in study design, data collection and interpretation, or the decision to submit the work for publication.

### Author contributions

PFYV, DD, Conception and design, Acquisition of data, Analysis and interpretation of data, Drafting or revising the article; YB, Acquisition of data, Analysis and interpretation of data; CP, Analysis and interpretation of data, Drafting or revising the article

### Ethics

Animal experimentation: This study was performed in accordance with the guidelines of the Animal Care Committee of the European Communities Council Directive (86/609/EEC) and were approved by the ethics committee of the University of Bordeaux (animal facility agreement 155 number C33-063-075)

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
