## [Decision Letter]

Thank you for submitting your work entitled "A synaptic F-actin network controls otoferlin-dependent exocytosis in auditory inner hair cells" for consideration by *eLife*. Your article has been reviewed by two peer reviewers, and the evaluation has been overseen by a Reviewing Editor and Gary Westbrook as the Senior Editor.

The reviewers have discussed the reviews with one another and the Reviewing editor has drafted this decision to help you prepare a revised submission. The reviewers thought that the work contains interesting new data that will be of interest to a wide audience of neuroscientists and cell biologists that justify publication in *eLife*. No new experiments are required, but we ask you to address their concerns related to experimental conditions (osmolarity), text editing and data analysis, which you will find below in the original comments.

*Reviewer #1:*

This is a very interesting paper with some surprising results. The role of actin in the process of exocytosis is unknown for auditory hair cell ribbon synapses, although these synapses have an abundant supply of actin, as shown in Figure 1 of this paper. At other synapses, it is thought that actin modulates the process of exocytosis and the recruitment of vesicles to the readily releasable pool (RRP). The authors of this paper show that a 45-minute treatment with latrunculin-A, a drug that depolymerizes actin, increases the amount of exocytosis in hair cells. Interestingly, it also appears to allow Ca^2+^ channels to be located further away from the synaptic ribbon. It thus appears that actin is involved in stabilizing the exact localization of Ca^2+^ channels at synaptic ribbon active zones. These are important and novel findings. They will be of interest to a wide audience of neuroscience and cell biology. I have only a few major concerns that need to be addressed:

1) In Figure 2 the authors should state whether the level of peak free Ca^2+^ concentration reached in both condition (control and latrunculin-A treated) was the same. If available, please show the Ca^2+^ concentration levels in panel 2C.

2) Another way to quantify the inflexion point (or breaking-point) is to calculate the first derivative of the traces in Figure 2 and then show a bar graph of the peak values of the first derivative. Is this what the authors have done? If not, I would like to see this also shown in panel 2D.

3) Changing the osmolarity of the internal solution 310 to 390 mOsm seems like a drastic change. Does this ever occur under physiological or pathological conditions? Please discuss this further.

4) Is there some particular physiological context where an osmolarity change from 310 to 390 can occur in the mammalian organ of Corti? Perhaps under pathological conditions? The authors need to discuss this more fully.

5) The increase in capacitance jumps with latrunculin-A seems to suggest that actin acts as a barrier that controls the trafficking of vesicles from the cytoplasm to the release sites near the synaptic ribbon. Electron-tomography evidence for actin and microtubules near the synaptic ribbon has been provided by Graydon et al. (JNeurosci., 2011; see their Figure 5B). This paper should be cited in the Introduction.

6) The level of free Ca^2+^ concentration reached in Figure 4 should be mentioned in the text or the figure caption. Average numbers should be given in the text for this set of experiments.

7) Please cite and discuss a recent paper that shows how changes in external hypertonic stress affect vesicle mobility (see "Ionic imbalance, in addition to molecular crowding, abates cytoskeletal dynamics and vesicle motility during hypertonic stress", by Nunes, P. et al. (PNAS, 2015). By contrast, the manipulations that the authors are performing may facilitate vesicle motility and thus increases in exocytosis.

*Reviewer #2:*

In the paper the authors present a fascinating series of observations linking an intracellular actin network to trafficking of synaptic vesicles, and potentially to changes in hydrostatic pressure. F-actin 'cages' appear to surround hair cell active zones. When cages are disrupted by latrunculin, the spacing of synaptic ribbons and calcium channel clusters is widened. Latrunculin treatment increases the voltage-evoked capacitance increase thought to reflect fusion of synaptic vesicles. Increased hydrostatic pressure also increased voltage-gated calcium current. Calcium-invoked capacitance increases are enhanced by increased hydrostatic pressure. This work raises many interesting questions. The initial observation of F-actin-dependent synaptic structure and function is significant and potentially quite important, particularly in view of the organized membranous domains of hair cells (see for example, Bullen et al., J Cell Sci. 2015 Jul 15;128(14):2529-40). A number of suggestions and questions follow:

1) Introduction, first paragraph: Introducing hair cells as 'highly sensitive barometers […]' in a paper that examines the effects of hydrostatic pressure might mislead many readers to think that the authors are suggesting that hair cells are stimulated by changes in hydrostatic pressure at acoustic frequencies. But of course the adequate stimulus is hair bundle displacement, not cytoplasmic pressure.

2) Results and Discussion, first paragraph – cages or cylinders? It’s not obvious from the images. Are these maximum intensity z-stack projections? We need more information in the figure legends.

3) Results and Discussion, second paragraph: Figure 1 need re-labeling.

4) Results and Discussion, second paragraph: Can the authors provide an estimate of the number of vesicles per ribbon that make up the RRP? Does it make sense?

5) Results and Discussion, second paragraph: "[…] a switch from nanodomain to microdomain […]." This seems like an unwarranted throw away statement in this context. The capacitance change being measured occurs over 100 ms, effectively steady-state with respect to buffer effects. EGTA concentration could affect the extent to which neighboring calcium sources interact and summate to produce a global effect on free calcium – is that what the authors intend? Without further explanation this reference to nano/micro is irrelevant.

6) Results and Discussion, second paragraph: The authors refer here to "number of docked vesicles at release sites", but later to "diffusion rate of the synaptic vesicles to the sites of release". These are quite different mechanisms to explain the effect of latrunculin, with the second idea probably closer to reality. What is the authors' conclusion?

7) Results and Discussion, third paragraph: ~1.2 pF capacitance increase with increased pressure, approximately equivalent to the calcium-evoked capacitance increase in control and latrunculin-treated cells (0.9 to 1.4 pF), but – 30-40-fold greater than RRP. So presumably not just synaptic vesicles – what other organelles could be involved in these much larger membrane fusions?

8) Results and Discussion, third paragraph: could the authors clarify the reference to 'mechanosensitive' Cav1.3 channels. Is it just that calcium channels are added to the membrane during the massive vesicular fusion caused by increased hydrostatic pressure?

9) In the Results and Discussion, third paragraph, please rephrase the following passage: “Notably, a change in Ca^2+^ current amplitude…”. The authors should replace “change” for “reduction”.

10) Results and Discussion, last paragraph: F-actin regulates vesicle trafficking by sensing and regulating focal changes in hydrostatic pressure? What are these 'focal changes in hydrostatic pressure'? How would these be generated?

11) Figure 1: provide hair cell outline in right hand panel at hi mag. Is this a z-axis projection? Also, panel B is mislabeled as panel C on figure. What are the large black holes at the bottom of latrunculin-treated IHCs? What are the dimensions of F-actin cages in latrunculin? Are these center-to-center distances of ribbon and CaV1.3? Please specify in the figure legend.

---

## [Author Response]

*The reviewers have discussed the reviews with one another and the Reviewing editor has drafted this decision to help you prepare a revised submission. The reviewers thought that the work contains interesting new data that will be of interest to a wide audience of neuroscientists and cell biologists that justify publication in eLife. No new experiments are required, but we ask you to address their concerns related to experimental conditions (osmolarity), text editing and data analysis, which you will find below in the original comments.*

Reviewer #1:

*[…] I have only a few major concerns that need to be addressed: 1) In Figure 2 the authors should state whether the level of peak free Ca^2+^ concentration reached in both condition (control and latrunculin-A treated) was the same. If available, please show the Ca^2+^ concentration levels in panel 2C.*

The levels of the peak intracellular Ca^2+^ concentration ([Ca^2+^]_free_) reached upon UV-flash Ca^2+^ uncaging were verified to be similar in control (n = 7) and latrunculin-treated IHCs (n = 5), respectively 59 ± 7 µM and 57 ± 5 µM (p = 0.9). These data are now added in Figure 2 (inset) and discussed in the fifth paragraph of the Results section.

*2) Another way to quantify the inflexion point (or breaking-point) is to calculate the first derivative of the traces in Figure 2 and then show a bar graph of the peak values of the first derivative. Is this what the authors have done? If not, I would like to see this also shown in panel 2D.*

As suggested by the reviewer, we analyzed the first derivative of the exocytotic curve of Figure 2 and plotted the results in Figure 2. Peak rate values, instead of breaking points, are now indicated as mean ± SEM in a bar graph (Figure 2, bottom right). Note that the analysis of additional cells bring the n values from 11 to 16 in controls and from 10 to 14 for latrunculin-A treated IHCs. This new data analysis is now discussed in the fifth paragraph of the Results section.

*3) Changing the osmolarity of the internal solution 310 to 390 mOsm seems like a drastic change. Does this ever occur under physiological or pathological conditions? Please discuss this further.*

*4) Is there some particular physiological context where an osmolarity change from 310 to 390 can occur in the mammalian organ of Corti? Perhaps under pathological conditions? The authors need to discuss this more fully.*

A discussion on the points raised by the reviewer is now in the eleventh paragraph of the Results section: ‘Do large changes in hydrostatic pressure occur in the cochlea during physiological or pathological conditions? […] Our study, showing that IHC exocytosis is sensitive to osmotic forces, suggests that defects in synaptic transmission at the auditory ribbon synapses may also underlie some hearing disorders associated with Menière's disease.’

*5) The increase in capacitance jumps with* latrunculin *-A seems to suggest that actin acts as a barrier that controls the trafficking of vesicles from the cytoplasm to the release sites near the synaptic ribbon. Electron-tomography evidence for actin and microtubules near the synaptic ribbon has been provided by Graydon et al. (JNeurosci., 2011; see their Figure 5B). This paper should be cited in the Introduction.*

This point has been added in the Introduction: **‘**Electron-tomography evidence for F-actin and microtubules near the synaptic ribbon has been observed in bullfrog hair cells (Graydon et al 2011).’

*6) The level of free Ca^2+^ concentration reached in Figure 4 should be mentioned in the text or the figure caption. Average numbers should be given in the text for this set of experiments.*

This point has been added: ‘The level of peak [Ca^2+^]_int_ reached upon UV-flash Ca^2+^ uncaging were verified to be similar in 310 mOsm/kg (n=7) and 390 mOsm/kg (n = 5) conditions, 59 ± 7 µM and 64 ± 6 µM (p = 0.6)’. These data are now added in Figure 4 (inset) and discussed in the ninth paragraph of the Results section.

*7) Please cite and discuss a recent paper that shows how changes in external hypertonic stress affect vesicle mobility (see "Ionic imbalance, in addition to molecular crowding, abates cytoskeletal dynamics and vesicle motility during hypertonic stress" by Nunes, P. et al. (PNAS, 2015). By contrast, the manipulations that the authors are performing may facilitate vesicle motility and thus increases in exocytosis.*

This recent paper is now discussed. It is now indicated in the tenth paragraph of the Results section: ‘In the same way, the mobility of vesicles in pancreatic cells and primary hepatocytes was shown to be affected by hydrostatic pressure, a process related to molecular crowding and microfilaments polymerization (Nunes et al., 2015).’

Reviewer #2:

[…] A number of suggestions and questions follow:

*1) Introduction, first paragraph: Introducing hair cells as 'highly sensitive barometers […]' in a paper that examines the effects of hydrostatic pressure might mislead many readers to think that the authors are suggesting that hair cells are stimulated by changes in hydrostatic pressure at acoustic frequencies. But of course the adequate stimulus is hair bundle displacement, not cytoplasmic pressure.*

This sentence has been changed in the Introduction and now reads: ‘Auditory hair cells convert tiny variations of sound pressure through the displacement of their apical hair bundles into analogous voltage waveforms.’

2) Results and Discussion, first paragraph – cages or cylinders? It’s not obvious from the images. Are these maximum intensity z-stack projections? We need more information in the figure legends.

The term cylinder is confusing and replaced by cage in the first paragraph of the Results and Discussion section. The confocal images shown in Figure 1 are averaged Z-stack projections of 20 (top left) to 8 (top right and bottom left) slices of 0.250 µm. By varying the depth of the Z-axis, a F-actin cage-like structure was observed surrounding the ribbon. These points are now added in the legend of Figure 1.

3) Results and Discussion, second paragraph: Figure 1 need re-labeling.

The labeling of Figure 1 has been modified.

*4) Results and Discussion, second paragraph: Can the authors provide an estimate of the number of vesicles per ribbon that make up the RRP? Does it make sense?*

We discussed this point in the second paragraph of the Results section (please see: ‘After 100 ms depolarization, from -80 to -10 mV, the exocytotic response in control IHCs reached a maximum amplitude of 22.0 ± 2.9 fF (n = 11). […] In bassoon mutants with abnormal number of anchored ribbons and reduced Ca^2+^ currents both short (20 ms) and long (100 ms) impulse activated-exocytosis were affected (Jing et al., 2013)’).

*5) Results and Discussion, second paragraph: "[…] a switch from nanodomain to microdomain […]." This seems like an unwarranted throw away statement in this context. The capacitance change being measured occurs over 100 ms, effectively steady-state with respect to buffer effects. EGTA concentration could affect the extent to which neighboring calcium sources interact and summate to produce a global effect on free calcium – is that what the authors intend? Without further explanation this reference to nano/micro is irrelevant.*

We agree with the reviewer. This section has been now modified to read: ‘This sensitivity to EGTA suggested a disorganization of the Ca^2+^ channel clusters in regards to the release sites in latrunculin-treated IHCs, in good agreement with confocal imaging (Figure 1). Rising the EGTA concentration could also affect the extent to which neighboring calcium sources interact and summate to produce a global effect on free calcium.’

*6) Results and Discussion, second paragraph: The authors refer here to "number of docked vesicles at release sites", but later to "diffusion rate of the synaptic vesicles to the sites of release". These are quite different mechanisms to explain the effect of latrunculin, with the second idea probably closer to reality. What is the authors' conclusion?*

The sentence has been modified (now in the Results and Discussion, fourth paragraph) to read: ‘Its disruption with depolymerising agents would therefore facilitate vesicle replenishment of the release sites, as shown in a large variety of secretory cells (Malacombe et al., 2006), by increasing the number of available vesicles for docking and priming. Alternatively, a disrupted F-actin could facilitate the diffusion of Ca^2+^ from its sites of entry and stimulate replenishment.’

*7) Results and Discussion, third paragraph: ~1.2 pF capacitance increase with increased pressure, approximately equivalent to the calcium-evoked capacitance increase in control and latrunculin-treated cells (0.9 to 1.4 pF), but – 30-40-fold greater than RRP. So presumably not just synaptic vesicles – what other organelles could be involved in these much larger membrane fusions?*

This point is now discussed in the sixth paragraph of the Results and Discussion section as follows: ‘This augmentation of the IHC resting membrane capacitance was about 50 times larger than the size of the RRP evoked by membrane depolarization (Figure 2; 22 fF). Where does this large addition of membrane come from? One explanation is that the high intracellular hydrostatic pressure triggered the fusion of a large amount of extrasynaptic vesicles to the plasma membrane as previously described during Ca^2+^ uncaging (Vincent et al., 2014).’*8) Results and Discussion, third paragraph: could the authors clarify the reference to 'mechanosensitive' Cav1.3 channels. Is it just that calcium channels are added to the membrane during the massive vesicular fusion caused by increased hydrostatic pressure?*

We agree with the reviewer, ‘we cannot exclude the addition of Ca^2+^ channels to the plasma membrane during the massive vesicular fusion caused by increased hydrostatic pressure. However, this mechanism appears unlikely in regards to results obtained in non-secretory cells such as smooth muscle and HEK cells where similar effect on Ca^2+^ channels were obtained (Lyford et al., 2002).’ This point is now added in the seventh paragraph of the Results and Discussion section.

9) In the Results and Discussion, third paragraph, please rephrase the following passage: “Notably, a change in Ca^2+^ current amplitude…”. The authors should replace “change” for “reduction”.

This is now indicated in the seventh paragraph of the Results and Discussion section (‘Notably, an increase in Ca^2+^ current amplitude in low bath hydrostatic pressure (equivalent to increasing intracellular pressure) was also reported in dissociated guinea pig vestibular hair cells (Duong Dinh et al., 2009; Haasler et al., 2009)’).

*10) Results and Discussion, last paragraph: F-actin regulates vesicle trafficking by sensing and regulating focal changes in hydrostatic pressure? What are these 'focal changes in hydrostatic pressure'? How would these be generated?*

To clarify this issue, this section has been rewritten to read: ‘In conclusion, we propose that a synaptic F-actin network tightly controls the flow of synaptic vesicles during exocytosis at the IHC ribbons. This synaptic F-actin network also influences the sensitivity of exocytosis to hydrostatic pressure changes, presumably through a regulation of membrane tension at the active zones.’

*11) Figure 1: provide hair cell outline in right hand panel at hi mag. Is this a z-axis projection?*

Done.

*Panel B mislabeled as panel C on figure.*

Done.

*What are the large black holes at the bottom of latrunculin-treated IHCs?*

‘The black holes at the base of the IHCs likely indicated swollen IHC active zones produced by the synaptic F-actin disorganization’. This point is now added in the legend of Figure 1.

*What are the dimensions of F-actin cages in latrunculin?*

The size of the synaptic F-actin cages in latrunculin treated IHCs could not be measured since they were completely disorganized (see statement in the first paragraph of the Results and Discussion section).

*Are these center-to-center distances of ribbon and CaV1.3? Please specify in the figure legend.*

We now specify in the legend of Figure 1 that these distances are the center mass distances.